# Influence of Face Masks on Physiological and Subjective Response during 130 Min of Simulated Light and Medium Physical Manual Work—An Explorative Study

**DOI:** 10.3390/healthcare11091308

**Published:** 2023-05-03

**Authors:** Benjamin Steinhilber, Robert Seibt, Julia Gabriel, Mona Bär, Ümütyaz Dilek, Adrian Brandt, Peter Martus, Monika A. Rieger

**Affiliations:** 1Institute of Occupational and Social Medicine and Health Services Research, Medical Faculty, University Hospital Tuebingen, 72074 Tuebingen, Germany; robert.seibt@med.uni-tuebingen.de (R.S.); julia.gabriel@med.uni-tuebingen.de (J.G.); mona.baer@med.uni-tuebingen.de (M.B.); uemuetyaz.dilek@daimler.com (Ü.D.); adrian.brandt@t-online.de (A.B.); monika.rieger@med.uni-tuebingen.de (M.A.R.); 2Institute for Clinical Epidemiology and Applied Biometry, Medical Faculty, University Hospital Tuebingen, 72076 Tuebingen, Germany; peter.martus@med.uni-tuebingen.de

**Keywords:** COVID-19, coronavirus, pandemic, respirator, cardiorespiratory fitness level, wearing comfort, human physiology, breathing system

## Abstract

Background: Undesirable side effects from wearing face masks during the ongoing COVID-19 pandemic continue to be discussed and pose a challenge to occupational health and safety when recommending safe application. Only few studies examined the effects of continuously wearing a face mask for more than one hour. Therefore, the influence of wearing a medical mask (MedMask) and a filtering facepiece class II respirator (FFP2) on the physiological and subjective outcomes in the course of 130 min of manual work was exploratively investigated. Physical work load and cardiorespiratory fitness levels were additionally considered as moderating factors. Methods: Twenty-four healthy subjects (12 females) from three different cardiorespiratory fitness levels each performed 130 min of simulated manual work with light and medium physical workload using either no mask, a MedMask or FFP2. Heart rate, transcutaneous oxygen and carbon dioxide partial pressure (P_tc_O_2_, P_tc_CO_2_) as well as perceived physical exertion and respiratory effort were assessed continuously at discrete time intervals. Wearing comfort of the masks were additionally rated after the working period. Results: There was no difference in time-dependent changes of physiological outcomes when using either a MedMask or a FFP2 compared to not wearing a mask. A stronger increase over time in perceived respiratory effort occurred when the face masks were worn, being more prominent for FFP2. Physical workload level and cardiorespiratory fitness level were no moderating factors and higher wearing comfort was rated for the MedMask. Conclusion: Our results suggest that using face masks during light and medium physical manual work does not induce detrimental side effects. Prolonged wearing episodes appeared to increase respiratory effort, but without affecting human physiology in a clinically relevant way.

## 1. Introduction

In 2019, the new coronavirus disease, causing the severe acute respiratory syndrome, emerged and became a global pandemic in 2020, with dramatically high fatalities in every country of the world. According to the World Health Organization (WHO), on 16 March 2023, 760,360,956 cases of COVID-19 and 6,873,477 related deaths were confirmed globally [1]. Additionally, personal lifestyles and social relationships were affected with significant implications for public health [2,3]. In 2023, the global COVID-19 pandemic is still underway, and protective measures against the spread of the responsible pathogen SARS-CoV-2 are continuously taken and adjusted. In this regard, face masks are an important measure to reduce the infection rate by droplet and airborne transmission [4]. Depending on the type of mask, their function is either self-protection or protection of others. Accordingly, medical masks (MedMask; European norm 14683) protect others to a very large extent if the wearer is infectious because exhaled droplets are not able to pass through the mask to the outside. Filtering facepiece respirators class II masks (FFP2; European norm 149, comparable to N95 and KN95 respirators [5]) are used to additionally protect oneself from infections since the mask material holds back the virus from the outside to a very large extent [6]. Depending on the setting, people were and are obliged by authorities to wear these masks in public and especially at work for longer periods of time. The effectiveness of face masks as a SARS-CoV-2 containment measure is undisputed and was confirmed by numerous scientific studies [7,8,9]. However, face masks are increasing users’ breathing resistance [10], leading to considerable public concern about possible adverse health effects for face mask users unrelated to the SARS-CoV-2 infection. These concerns are fueled by some publications describing that wearing face masks might have negative health effects [11,12] and opinions released on the Internet (e.g., [13]). Their main argument was that the breathing system will be altered by the masks leading to hypoxemia and hypercapnia with detrimental long-term health effects, although the majority of scientific literature indicates no detrimental effect on vital parameters in healthy persons [14,15].

Since occupational safety and health are responsible for preventing health hazards in the workplace, such concerns must be taken seriously. Therefore, in an early phase of the pandemic, the German Social Accident Insurance recommended that face masks should not be worn for more than 2 h at a time, with a subsequent break in wearing for 30 min [16] without the support of empirical data. Most of the published studies on potential physiological impairments by using facemasks had observation periods of less than 1 h [17] and observed physical activities with rather limited relevance to everyday life and real occupational work [15,18].

As occupational health and safety authorities need to make recommendations for the wearing of face masks in the workplace with special consideration of possible adverse effects on face mask users’ health, further studies are needed with longer wearing episodes, especially exceeding the duration of 2 h since this has been proposed as maximum continuous wearing duration [16], as well as more work-related study scenarios.

Prior to the pandemic, the question of possible negative side effects from wearing face masks was of minor interest to the scientific community and many questions, especially about possible moderating factors, remain unanswered. In this respect, it is possible that the side effects of the masks depend on the level of cardiovascular stress or physical workload. For example, Fikenzer et al. [19] showed impairments in pulmonary function parameters and performance under maximum load conditions during an incremental bicycle ergometer test when using a MedMask and FFP2 mask, whereas in a study by Georgi et al. [20] and in our own previous study [21], only non-relevant or no effects of using masks on physiological parameters occurred during submaximal bicycle ergometry. Furthermore, it is presumed that the individual cardio-respiratory fitness level may moderate the effects of using face masks [20].

In this respect, we exploratively investigated the influence of wearing a MedMask and a FFP2 mask on physiological outcomes associated with cardio-respiratory demands as well as perceived physical exertion and respiratory effort in the course of 130 min continuous manual work. In addition, the importance of two potential moderating factors (physical workload and cardiorespiratory fitness) was examined and the comfort of wearing the two face masks was assessed complementarily.

## 2. Materials and Methods

### 2.1. Participants

Twenty-four healthy persons (12 females, 12 males) participated in this study. They were recruited by email announcements within the University of Tübingen and the University Hospital of Tübingen as well as mouth to mouth propaganda. Exclusion criteria included metabolic diseases such as diabetes mellitus, cardiovascular or respiratory diseases, and pregnancy. Additionally, a medically unremarkable pulmonary function test (spirometry) and electrocardiogram during a bicycle ergometer test until exhaustion assessed by a physician were further requirements for inclusion. The full list of in- and exclusion criteria corresponding to the German guideline for ergometry within occupational medical examinations [22] can be found in Appendix A. The results of the initial bicycle ergometer test were further used to determine participants’ individual maximal physical working capacity (PWCmax) as an indicator of cardiorespiratory fitness [23]. PWCmax is the maximum mechanical power in Watt per kilogram body weight (W/kg) reached at the very end of the bicycle ergometer test. According to published PWCmax norm values [24], participants were categorized into three sex-specific cardio-respiratory fitness levels: PWCmax below the norm (men < 3.0 W/kg, women < 2.6 W/kg), corresponding to the norm (men 3.0 W/kg ≤ x < 4.1 W/kg, women: 2.6 W/kg ≤ x < 3.5 W/kg), and above the norm (men: ≥4.1 W/kg, women: ≥3.6 W/kg). The study was approved by the Ethics Committee of the Medical Faculty of the University of Tübingen, and written informed consent was provided by each participant prior to participation. After study completion, the participants received financial compensation. The study is part two of a research project which was registered in the German clinical trial register (DRKS00024531).

### 2.2. Study Design and Study Procedure

#### 2.2.1. Study Design and Sample Size

A randomized, within-subject design was used for comparing three experimental conditions using either a MedMask, a FFP2, or no mask (control) when performing 130 min of simulated manual work with light and medium physical workload in a laboratory environment (Figure 1). As mentioned in the introduction, the effect of face masks may depend on the physical workload, which is why the simulated manual work was performed once as light physical work and once as medium physical work. This simulation was, thus, representative for a substantial portion of the tasks performed by employees in the manufacturing industry, which in Germany comprises 8.11 million employees [25]. The duration of 130 min was chosen to exceed an uninterrupted wearing time of two hours, which was suggested as the maximum wearing time during the early phase of the pandemic [16].

The sample size was determined for avoiding first order carry over effects with respect to the number of the three main experimental conditions. Therefore, a Williams design for an uneven number of treatments (three experimental mask conditions), which combined two Latin Squares with 6 order sequences [26], was applied and the sample size was set to 24 as a multiple of 6 with 4 subjects per order sequence. The order of physical workload (light or medium) was held constant on each experimental day but was balanced and randomized in a way that half of the subjects started each day with the light physical workload and the other half with the medium physical workload. Each subject, which was confirmed for inclusion after the initial day, participated in all 3 experimental days and all 6 experimental conditions with the exact time schedule, as shown in Figure 1.

#### 2.2.2. Face Masks

Over the course of the pandemic, recommendations for wearing face masks were constantly adjusted to reflect the prevailing infection situation and burden on health care systems. Within this process, the MedMask and respirators such as FFP2 without exhalation valve emerged as the most important ones. Therefore, a disposable medical face mask without exhaling valve (NITRAS Medical Care Dental GmbH, 4331//PROTECT, medical face mask, made of fiberglass-free non-woven fabric, blue, 3-ply, integrated nosepiece, round and latex-free elastic bands, manufactured according to EN 14683 Type IIRv) and disposable filtering face piece respirator without exhalation valve, protection class II (Honeywell PREMIUM, 5000 series, model 5210, EN 149) were used in the present study. Currently, the WHO (January 2023) recommends MedMasks for the public and respirators such as the FFP2 mask are recommended for caregivers providing care to suspected or confirmed COVID-19 patients [27].

#### 2.2.3. Simulated Light and Medium Physical Manual Work

The work simulation environment was based on containers and could be set up to perform a manual task with light or medium physical workload by changing the loads to be manipulated (Figure 2). In detail, we used beverage crates containing common 1-L plastic bottles which were either empty (light workload) or filled with water or a sand-water mixture to realize loads of 1 or 2 kg (medium workload), respectively. Handling the complete beverage crate resulted in 10 to 15 kg in case of filled bottles. The task lasted 130 min in order to exceed the wearing limit of two hours recommended in the early stage of the pandemic by the German Accident Insurance [16] and consisted of three subtasks with a focus on handling the bottles. Light and medium physical manual work was classified according to the REFA system [28] with the weights to be handled continuously (0–1 kg light and 1–3 kg medium physical workload) serving as the basis for the work simulation. In the case of the light physical manual task, the bottles to be handled were empty, and in case of the medium physical manual task, the bottles were filled with sand and water. In order to ensure a similar work pace among the participants, a bar moving from left to right was shown on two screens above the simulated work environment (Figure 2). When the bar reached the right end of the screen, the corresponding subtask had to be completed. Participants were instructed to think of the bar as a temporal guide to maintain a constant work pace during each subtask. They were reminded to adjust their work pace if necessary but had to complete the subtask even if it took a bit longer.

Three subtasks were performed. Subtask one (placing bottles) consisted of placing bottles with colored lids according to a given pattern one by one into three areas of a height adjustable wall-system made of beverage crates. For individual height adjustment, the rim of the upper beverage crates of the wall-system was aligned to the extended arm of the subjects in an upright standing posture with shoulder anteversion angle of 90°. This ensured that about two third of the work was carried out below shoulder height and approximately one third above. The walls consisting of three areas (left, middle, and right) and this subtask lasted 5 min and 15 s. During subtask two (carrying beverage crates), participants had to place bottles with colored lids (red, blue, green) into two beverage crates located on the floor in front of the wall according to a given color pattern and move them from one side of the wall-system to the other within 1 min and 30 s. Subtask three (placing clothespins) included placing 24 colored clothespins one by one onto the rims of the beverage crates of the middle wall according to a given color pattern within 1 min and 30 s. Details of the three subtasks are given in Appendix B.

#### 2.2.4. Procedures

The participants visited the laboratory four times. On their initial visit, they were informed about the study procedures and then signed the informed consent. In- and exclusion criteria were verified including the pulmonary function test before they were familiarized with the physical manual work task for around 22 min. Thereafter, parts of the German version of the Nordic Questionnaire [29] and of the Physical Activity, Exercise, and Sport Questionnaire [30] were administered to determine sex, state of smoking, age, body mass index (BMI), and physical sports activity. Finally, the bicycle ergometer test took place under medical observation to verify the final study admission in case of a medically unremarkable ECG and to assign the subjects to one of the three fitness groups.

Each experimental day began by preparing the participant with the measurement equipment and instructions on how to use the assigned face mask, followed by a 20 min resting period of supine lying with the lower legs elevated. According to the randomized experimental mask condition, the MedMask, FFP2, or no mask was then donned before the first period of 130 min light or moderate physical manual work took place. After a 35 min rest break (without face mask, therefrom the last 20 min supine lying), participants had to work for another 130 min again with light or moderate physical manual work. Half of the participants started each day with light physical work, the other half in the reverse sequence. The assignment to the applied sequence was randomized. Within the working periods, the three subtasks were accomplished in a consecutive order. Subtask 1 was dominant and alternated either with subtask 2 or 3. Concretely, after completing subtask 1, participants performed subtask 2, then subtask 1 again, followed by subtask 3, and then, everything from the beginning for a duration of 130 min.

Participants’ level of perceived physical exertion and respiratory effort were gathered during 45 s of quite standing. These assessments took place, directly before each working period while already wearing the mask corresponding to the experimental condition and every 26 min during the two 130 min working periods. The rater marked these periods in the data file by pushing a marker button on the keyboard of a laptop that was connected to the measurement device for recording heart rate via Bluetooth. Additionally, the rater paused the work pace timer (moving bar on the screen) during each of these short interruptions.

In-between the two 130 min work period, there was a recovery break without mask wearing. During the first 15 min of the 35 min rest break, participants were provided a standard meal (sandwich and fruit) and the rest room could be used. The last 20 min of the break again consisted of supine lying with the lower legs elevated like prior to the first work period in the beginning of each experimental day. There were at least 2 days between the initial day and the first experimental day. The time between the three experimental days were 2 to 7 days and all experimental days were conducted at the same time of the day.

### 2.3. Outcomes and Measuremsents

#### 2.3.1. Heart Rate

Heart rate was recorded continuously using electrocardiography (ECG) at a sampling rate of 1000 Hz (PS12 device, THUMEDI^®^ GmbH & Co. KG, Thum, Germany, CE certified medical device). Before placing the electrodes, the skin was shaved in case of breast hair and prepared with an abrasive paste (Skin Prep Gel, Nuprep^®^, Aurora, CO, USA). A ground (neutral) electrode was placed over the cervical vertebra C7 and two Ag/AgCl surface electrodes (42 × 24 mm, Kendall^TM^ H93SG ECG Electrodes, COVIDIEN, Zaltbommel, the Netherlands) were placed on participants’ chest (electrode one: ~5 cm cranial and ~3 cm left lateral from the distal end of the sternum; electrode two: at the level of the fifth left rib between the anterior and mid axillary line) for heart rate measurement.

#### 2.3.2. Transcutaneous Oxygen and Carbon Dioxide Partial Pressure

A transcutaneous gas monitor (IntelliVue TcG10, Philips Medical Systems DMC GmbH, Boeblingen, Germany, connected to a transcutanuous sensor 84, Radiometer GmbH, Krefeld, Germany) with a sampling rate of approximately 0.125 Hz was used for noninvasive measurement of oxygen and carbon dioxide partial pressure. The sensor was placed on the right upper arm over the middle deltoid muscle, with an adhesive ring and a drop of contact fluid between skin and sensor at the beginning of the first 20 min rest period. After about 10 min, once the measurement site was warmed up by the sensor to a temperature of 44° Celsius, the measurement data stabilized. Measurement of P_tc_O_2_ and P_tc_CO_2_ were previously shown to provide reliable data even during physical activity [31].

#### 2.3.3. Subjective Measurements

Perceived respiratory effort and physical exertion were assessed by using a modified Borg CR10 scale (0 = nothing at all, 10 = maximal) every 26 min during the 130 min of simulated work. Typically, the scale is used to assess the perceived physical exertion, but it is also used to assess perceived respiratory effort during exercise [32].

#### 2.3.4. Survey on Mask Wearing Comfort

Wearing comfort was assessed at the very end of the two experimental days on which the masks were worn. Therefore, a questionnaire proposed by Li et al. [33] for assessing discomfort due to mask wearing was handed to the subjects. The questionnaire included 10 different sensations of discomfort which were operationalized by the following terms: humid, hot, breathing resistance, itchy, tight, salty, unfit, odor, fatigue, and overall discomfort. These terms were translated to German and two discomfort sensations (headache and dizziness) were added. Each discomfort sensation was rated using a scale ranging from 0 to 10 with (0.5 increments) by the participants. Participants were provided additional orientation when using the scales. A level of zero was connected to “not at all”, 5 to “mildly” and 10 to “strongly”. In the case of the “overall discomfort” sensation, the designation of the 0 to 10 sale was 0 representing “comfortable”, 5 “uncomfortable”, and 10 “extremely uncomfortable”. Furthermore, the scales were divided into three equidistant ranges with separators after 3 and directly before 7 [33]. There are no data available on the quality criteria of this assessment tool. However, due to the lack of a validated instrument for assessing wearing comfort of face masks and in terms of comparability to published data, we decided to also use this instrument.

### 2.4. Data Analysis

#### 2.4.1. Parameter Calculation

In order to be able to analyze changes over time in the objective parameters (heart rate, P_tc_O_2_, and P_tc_CO_2_), the 130 min work periods were divided into 6 successive phases. The pre phase was defined as 5 min before the start of the physical work task while participants were still resting (lying) without mask. The following 130 min physical work period was paused every 26 min for 45 s for assessing the level of physical exertion and respiratory effort. This resulted in five phases of uninterrupted continuous work of about 25 min. Individual median heart rate and median transcutaneous partial pressures (P_tc_O_2_ and P_tc_CO_2_) were calculated for each of these phases. For analyzing time changes in perceived physical exertion and respiratory effort, the values obtained from six consecutive ratings every 26 min were used.

#### 2.4.2. Data Synchronization

In order to synchronize the physiological data, the internal quartz clocks of the two measuring instruments (PS12 device and transcutaneous gas monitor) were aligned by noting the exact times before and after each measurement. Due to the two independent measuring systems without a common data logger interface, the data naturally did not match to the second, even when referring to the internal clocks. However, with regard to the intended data analysis, with phases of about 25 min, we considered this to be negligible. This was even so for the 5 min pre-phases before the work periods, especially because the test persons already spent several minutes in the lying position before.

#### 2.4.3. Statistical Analysis

The pre-phase values (before the beginning of the working period and without face mask) of heart rate, P_tc_O_2_, and P_tc_CO_2_ on each experiment were compared to ensure similar pre-conditions by applying a one-factorial analysis of variance.

The major objective of this exploratory study was to investigate potential effects of the two face masks on temporal changes in physiological and subjective strain parameters throughout a 130 min physical work period. In addition, we explored whether physical workload and cardiovascular fitness level may be moderating factors in this respect. Therefore, linear mixed models (LMM) with the independent variables mask condition (3 levels), time (note: heart rate, P_tc_O_2_, and P_tc_CO_2_ had only 5 levels since participants were not wearing masks during the pre phases; perceived exertion and respiratory effort had 6 levels), workload (2 levels), and cardiorespiratory fitness (3 levels) including interaction terms (two-fold interaction: mask condition with time; three-fold interactions: mask condition with time and cardiorespiratory fitness level, mask condition with time and workload) on the dependent variables were applied. The alpha level of the LMM analyses was Bonferroni adjusted for 5 comparisons and set to alpha = 0.01. Furthermore, we used Tukey’s honest significant difference test for post hoc comparison which accounts for multiple comparison.

For the results of the survey on mask wearing comfort, the intraindividual difference between the two types of masks was calculated for each of the 12 items. Then, these differences were tested for deviations from zero using t-tests. Due to twelve comparisons, the alpha level for this analysis was adjusted according to Bonferroni, leading to statistically significant findings when the *p*-value was smaller than 0.004.

The variables for characterizing the study population are presented by descriptive statistics using mean and standard deviations or absolute and relative frequencies. However, differences among the three subgroups representing the different cardiorespiratory fitness levels were analyzed using a one factorial analysis of variance or chi-square tests according to the scaling of the outcome variable. The statistical software JMP^®^ 16.2.0 (SAS Institute Inc., Cary, NC, USA) was used for statistical analyses.

## 3. Results

### 3.1. Dropouts

The goal was to recruit approximately the same number of study participants in each fitness level. Subsequently, two participants had to be excluded after the initial visit because they did not meet the fitness level still required. Two more participants refused to participate after the first experimental day. In one case, the reason was migraine on the day after the first experimental day and the assumption that this would also occur after the next two experimental days. The second person refused further participation for unknown reasons.

### 3.2. Characteristics of the Final Study Sample

Participants (12 men and 12 women) were 37.8 ± 13 years old, had a mean BMI of 23.8 ± 2.3 kg/m², and only one participant reported being a smoker. On average, participants had weekly physical sports activity of 132.6 min. The final study sample included nine participants with a cardio-respiratory fitness level below the norm, ten corresponding to the norm, and five above the norm. In this respect, there were statistically significant differences in age, BMI, amount of weekly physical sport activity, and PWCmax between these subgroups. Table 1 provides a detailed overview of the characteristics of the final study sample.

### 3.3. Normal Distribution and Missing Data

The outcome variables were visually inspected and an acceptable level of normal distribution based on histograms, skewness and kurtosis (between −1 and 1) could be assumed. Furthermore, the applied statistical analyses (linear mixed models) are considered being robust against violation of normal distribution [34]. There were no missing data for perceived respiratory exertion or physical effort as well as the wearing comfort survey. Due to software problems, some missing data occurred for the outcomes heart rate (4.3%) and transcutaneous partial pressures of O_2_ and CO_2_ (6.4%).

### 3.4. Physiological Outcomes—Heart Rate, Transcutaneous O_2_ and CO_2_

First, the baseline values of physiological outcomes from each experimental condition are presented. Thereafter, changes over time in physiological outcomes when using face masks will be reported, as well as significant main effects related to mask use.

#### 3.4.1. Baseline Values of Heart Rate, Transcutaneous O_2_ and CO_2_ Partial Pressure

There were no statistically significant differences in baseline heart rate, transcutaneous O_2_, and CO_2_ partial pressure on the three experimental conditions while participants were not wearing a face mask prior to the working period (Table 2).

#### 3.4.2. Heart Rate

There was no statistically significant influence of using any of the face masks on the heart rate progression nor a moderating effect of fitness level or physical workload indicated by non-significant interactions of face mask with time or face mask with time and fitness level or workload (Figure 3, Table 3). A main effect was found for mask condition with a slightly elevated heart rate of 4 bpm for the overall 130 min work period when using the FFP2 compared to the other two mask conditions with *p*-values less than 0.001 shown by the post hoc comparison (no mask = 91.0 ± 17.0 bpm; MedMask = 91.6 ± 17.8 bpm; FFP2 = 95.2 ± 19.5 bpm). There was no difference between MedMask and control.

#### 3.4.3. Transcutaneous O_2_ and CO_2_ Partial Pressure

The progression of P_tc_O_2_ and P_tc_CO_2_ was not influenced by the applied mask. Again, fitness level and physical workload were no moderating factors (Table 3). In Figure 4, both outcomes are shown in the time course separately for physical work severities and mask conditions (Figure 4a–d).

In both transcutaneous partial pressure outcomes, the main effect of mask condition did not reach statistical significance after adjusting for multiple comparison. However, in tendency, P_tc_O_2_ was somewhat lower when using the MedMask compared to the control (no mask = 75.9 ± 9.4 mmHg, MedMask = 73.6 ± 9.6 mmHg, FFP2 = 75.1 ± 18.1 mmHg) and P_tc_CO_2_ mask was slightly lower when using FFP2 compared to control (no mask = 34.8 ± 4.4 mmHg, MedMask = 35.0 ± 3.5 mmHg, FFP2 = 34.5 ± 4.8 mmHg).

### 3.5. Subjective Outcomes—Perceived Physical Exertion and Respiratory Fitness Level

#### 3.5.1. Perceived Physical Exertion

Although perceived physical exertion level increased over time, these changes were similar among mask conditions and were not influenced by fitness level or workload (Table 4, Figure 5a,b). Irrespective of temporal changes, the ratings of perceived physical exertion were at a low level but were slightly higher when face masks were worn (no mask: mean = 1.2 ± 1.1, MedMask = 1.4 ± 1.3, FFP2 = 1.7 ± 1.4; Borg CR10 scale 10 = maximum). These elevations were statistically significant for FFP2 compared to both other experimental conditions (*p* < 0.0001) and also when using the MedMask compared to the control condition (*p* < 0.007).

#### 3.5.2. Perceived Respiratory Effort

Perceived respiratory effort was the only outcome where the applied mask had an effect on the time course. The statistically significant two-fold interaction of mask condition with time indicated a stronger increase in respiratory effort when using the face masks compared to the control condition which was more prominent when using FFP2 (Figure 5c,d and Figure 6). Post hoc comparison revealed statistically significant elevated values when using FFP2 already after 26 min and remained significantly elevated for all other time points compared to not using a mask and the MedMask condition as well. In the MedMask condition, respiratory effort also increased over time but was less intense, leading to statistically significant higher ratings for the time points 78 min, 104 min, and 130 min, compared to not using a mask. In general, the level of respiratory effort was rather low, even at the end of the 130 min working period, the median and 75-percentile level was about 3 and 4 on the 0–10 scale, respectively (Figure 5d).

Within the three experimental conditions, there was no further increase in perceived respiratory effort over time during last three evaluations (78, 104, and 130 min) for both face masks conditions, indicating a kind of steady state condition from 56 min of the work period. Under the control condition with no mask, this steady state was reached earlier already after 26 min, since there was no further increase in respiratory effort within the last four assessments.

The three-fold interactions were not statistically significant, giving reason to assume no moderating effect of fitness level or workload. The boxplots of perceived respiratory effort over the time course is given in Figure 5c,d, separately for workload and mask condition.

#### 3.5.3. Wearing Comfort

Wearing the MedMask was rated to be more comfortable than the FFP2 mask. The ratings were found to be statistically significant for the discomfort sensations heat, breathe resistance, itchy, and tightness (Figure 7). Additionally, the overall discomfort score was statistically significantly lower when using the MedMask. The relative frequency of “mildly” ratings (>3 and <7) reached 33% in the FFP2 mask condition and 22% in the MedMask condition. For ratings related to strongly (>7), the relative frequency was 33% and 6% for the FFP2 and MedMask, respectively. Additionally, in the MedMask condition, the overall discomfort scores were statistically significant lower (MedMask: Median 2.5, interquartile range 1.8, FFP2: Median 5.3, interquartile range 2.5) and did not correspond to the values for “uncomfortable” (scale value 5) or “extremely uncomfortable” (scale value 10).

## 4. Discussion

The results of the present study indicate no change in physiological outcomes in the time course of 130 min of continuous physical manual work when using either a MedMask or a FFP2 compared to the control condition without mask. A stronger increase over time in perceived respiratory effort was found when the face masks were worn, being more prominent for the FFP2. However, physical workload level and cardiorespiratory fitness level were no moderating factors in the context of these results. Finally, a higher wearing comfort was rated for the MedMask by the participants.

### 4.1. Physiological Outcomes—Heart Rate, P_tc_O_2_, and P_tc_CO_2_

The main finding of our study indicates that human physiology in healthy persons may not be affected in a clinically relevant manner by using face masks during longer wearing episodes. This is in line with two recent reviews summarizing the effects of face masks related to short terms applications (most studies included protocols with less than 60 min mask wearing) in healthy participants [15,17]. In addition, the results of Rebmann et al. [35], a study that was conducted before the pandemic, support this interpretation. They performed a longitudinal analysis of several physiological outcomes in 10 nurses using a N95 masks (similar to FFP2) or a N95 with surgical mask overlay in the time course of a 12 h work shift. While blood pressure, O_2_ levels, and heart rate did not change over time, there was a statistically significant increase in transcutaneous measured CO_2_ levels over time (beginning to end of the shift) which was more prominent when using N95 with surgical mask overlay. However, these changes did not exceed clinically relevant levels and the authors concluded that, from a physiologic perspective, the long-term application of N95 respirators does not cause negative effects.

In addition to our analysis regarding temporal changes, a statistically significant main effect of mask condition was found in heart rate with a slight elevation of 4 bpm when using FFP2 compared to control. Using the MedMask did not lead to an overall higher heart rate than without the mask. This is in line with the results of many previously published studies, which indicated that FFP2 (or similar respirators such as N95) has greater effects than MedMask, though the effects are also clinically non-relevant [15]. In contrast, Lässing and colleague found a 5 bpm higher heart rate when using a surgical face masks compared to not using a mask during a 30 min constant bicycle ergometer test at the individual maximal lactate steady state. Compared to our study, the applied physical exercise protocol was physically more demanding which could contribute to different effects of the surgical masks since it was suggested that the effects of surgical masks are more pronounced during severe exercise performance [36]. Furthermore, and probably more relevant, their methodology of assessing pulmonary and metabolic parameter required wearing a rubber mask over the surgical mask during the experimental condition. Sealing the face mask with a rubber mask was already mentioned as a limitation by the studies evaluating face masks using open-circuit spirometry systems [19,36,37]. In addition, a methodological paper of our group investigated the bias induced by face mask evaluation with additional respiratory measurement masks. It could be demonstrated that sealing the face mask by additional respiratory measurement masks significantly increases breathing resistance compared to normal use and leads to an overestimating of face mask effects [38].

The reason why we found a small general increase in heart rate in the FFP2 condition is likely to be the result of higher respiratory work required to overcome an increased breathing resistance [15]. However, from a work physiology perspective, a 4 bpm heart rate elevation can be regarded as not clinically relevant, especially since the median heart rates within all three experimental conditions remained below the proposed endurance limit of 105–110 bpm. According to the German guideline for the application of heart rate and heart rate variability in occupational medicine and occupational science, the endurance limit represents the physical exertion which characterizes the maximum muscular work that can be maintained over a regular working shift of 8 h [39].

The main effects of the mask condition tended to become statistically significant in P_tc_O_2_ and P_tc_CO_2_. The findings did not reach statistical significance since the alpha level was adjusted for multiple comparisons. However, several other studies reported some effects of mask wearing on transcutaneous partial pressure such as P_tc_CO_2_. Georgi et al. [20] found statistically elevated P_tc_CO_2_ levels when using a fabric mask, MedMask, or FFP2 during 3 min cycling at 50, 75, and 100 Watt on a bicycle ergometer, and Rebmann et al. [35] revealed an increase in P_tc_CO_2_ in nurses within a 12 h work shift. However, none of these statistically significant findings exceeded normal ranges and were interpreted as not clinically relevant by the authors. Additionally, in the present study, P_tc_O_2_ and P_tc_CO_2_ were within normal ranges for healthy adults: P_tc_O_2_ ≥ 60.0 mmHg [40], P_tc_CO_2_ ≤ 45 mmHg [41].

### 4.2. Perceived Physical Exertion and Respiratory Effort

The progression of perceived physical exertion was not rated differently when the face masks were used but ratings were generally somewhat higher. Respiratory effort increased more when face masks were worn. This increase was pronounced when FFP2 was used with a more rapid increase and a higher mean level of respiratory effort. A recent systematic review and meta analysis [17] on the effects of wearing face masks during exercise also found elevated physical exertion levels and higher levels of dyspnea when face masks were used. Lee et al. [10] showed that wearing a N95 respirator significantly increased inspiratory and expiratory flow resistances and lead to a reduction in air exchange volume, which requires more respiratory work for the same physical performance. Further investigations clarifying whether regular N95 respirator use may induce abnormal respiratory ability or function were recommended. However, in view of the physiological outcomes of the present study and conclusions of two recent reviews [15,17], it seems that the slightly increased respiratory work to overcome the breathing resistance was rather not accompanied by any negative consequences.

### 4.3. Wearing Comfort

Higher wearing comfort was rated for the MedMask in all of the 12 items compared to FFP2. In five ratings, the differences reached statistical significance. These findings are in line with the results reported by Li et al. [33], who found less discomfort sensation while wearing a surgical mask compared to N95 mask using the same questionnaire as in the present study. In their study, ten healthy participants performed one 20 min and two 10 min walking periods with different velocities on a treadmill in a climate chamber (25 °C, humidity of 70%) and were asked to repeatedly rate their perceived discomfort. In this regard, a general increase in discomfort sensation over time while wearing a face mask was found with no interaction between wearing duration and the applied face mask. Ratings were generally lower than in the present study, which may be partially explained by the shorter wearing duration in the study by [33] and potentially by differences in mask design of the applied face masks, regardless of the protection class, since mask design was shown to be an independent factor also influencing wearing comfort [42]. In a study by Radonovich et al. [43], the most frequent reason for discontinuing mask use before the end of an 8 h shift in health care workers was heat development followed by pressure or pain. With respect to the results of the present study, where clear differences were found between the MedMask and the FFP2 for heat generation and perceived tightness, this would indicate lower compliance for the FFP2.

### 4.4. Recommendation of Face Mask Use

When developing recommendations for using face masks for infection prevention at the workplace in the context of occupational health and safety, the consideration of possible harmful side effects was always an important issue. In Germany, early in the course of the pandemic, general recommendations for the use of respirators in dusty environments were transferred one-to-one to the use of coronavirus face masks, which allowed wearing any face mask for no longer than two hours [16]. These general recommendations have since been updated. By now, the recommendation on the use of respiratory protection at the workplace specifically mention that they do only apply to particle-filtering masks worn in dusty environments and not for face masks for infection prevention related to the COVID-19 pandemic [44]. Overall, there is a lack of research data, particularly on the physiological effects of face masks on the users. The data presented here suggest that prolonged continuous use is associated with no physiologically justifiable health risks, although respiratory effort increases. Other significant aspects, which are beyond the scope of the present study, may be related to the hygiene of the face masks. A study by Buzzer et al. suggested that regular replacement of soaked masks may be more important than wearing time limits. The authors reported that using a disposable face mask for several hours, without proper disposal or cleansing, may lead to collecting, growing, and cumulating inorganic as well as organic matter [45].

### 4.5. Strength and Limitations

A strength of the present study was the long observation period of 130 min, which was longer than in most other recent studies investigating the physiological response of wearing face masks [15,17]. Although this duration is still not representative for a full 8 h working day, it exceeds the recommended duration of continuous face mask wearing in a dusty environment at the workplace in Germany [44] and enables to discuss its physiological basis regarding wear time limits of face masks for infection prevention. Furthermore, the applied physical activity was a simulation of manual work, which better reflects the common physical requirements of workers in the industrial sector than standardized bicycle of treadmill protocols which still form the basis of our knowledge on the effects of face masks during physical occupational work [19,20,33,37]. Finally, our methodological approach did not change the leakage of the applied face masks.

A limitation of the present study was the sample, which was small in size and only included healthy participants. Additionally, although we did not use a convenience sample of young students, but rather a broad age composition of working-age people with different levels of cardio-respiratory fitness levels and applied a within-subject design, a selection bias cannot be ruled out, in the sense that participants with a positive attitude towards using face masks were more likely to participate in this study. The work simulation also had some limitations, especially regarding the two applied physical workloads (light and medium workload). The workloads were determined according to REFA [28], a common method for the design of work systems. This determination of work load did not necessarily match with the physical activity level classification based on energy consumption according to metabolic equivalents. Since energy consumption was not measured in the present study, the heart rate analyses provide the best indication of whether the two work loads could be differentiated metabolically. The mean heart rate was 86 ± 16 bpm for the light physical workload and 95 ± 19 bpm for the medium physical workload. Expressed as relative values related to the maximum heart rate (HRmax) measured during the initial bicycle ergometer test, light physical workload was characterized by 49 ± 9% HRmax and medium physical workload by 54 ± 11% HRmax. These relative heart rates indicate that the difference in physical activity was rather small and both workloads were in a low to very low intensity range [46]. Higher intensities such as those found in heavy physical work and heavy labor were not considered. Therefore, it could be possible that a moderating effect of workload was not detected due to a too narrow range of the investigated workloads. Other factors with a potential influence on breathing physiology, such as the menstrual cycle in women [47], were not monitored or considered in the present study. Furthermore, we only focused on potential side-effects due to mask wearing associated with the breathing system, although the masks may alter additional parts of the human body such as the muscular system of the masticatory muscles. Changes in resting muscle activation of masticatory muscles were discussed as a potential factor of headache related to prolonged mask use [48,49]. Finally, our set of physiological measurements was limited to heart rate and P_tc_O_2_ and P_tc_CO_2_ only and blood gases may be more accurate by the analysis of blood samples [50].

## 5. Conclusions

From a physiological point of view, our data suggest that surgical masks and filtering facepiece respirator face masks can be worn during light and moderate physical manual work for extended periods of time without health hazards in healthy workers. Physical workload level and cardiorespiratory fitness level further seem not to moderate the physiological effects of face masks in the time course. Some of the subjective outcomes (respiratory effort and wearing comfort) indicate potentials for psychological stress when wearing face masks, especially for the application of the FFP2. Although the majority of published studies, as well as the results of the present work, so far give little reason to believe that there is a relevant health risk from wearing face masks for COVID-19 infection prevention, further empirical data on the effect of the masks on vital signs over the duration of an entire work shift would help solve remaining issues for occupational health and safety.

## Figures and Tables

**Figure 1 healthcare-11-01308-f001:**
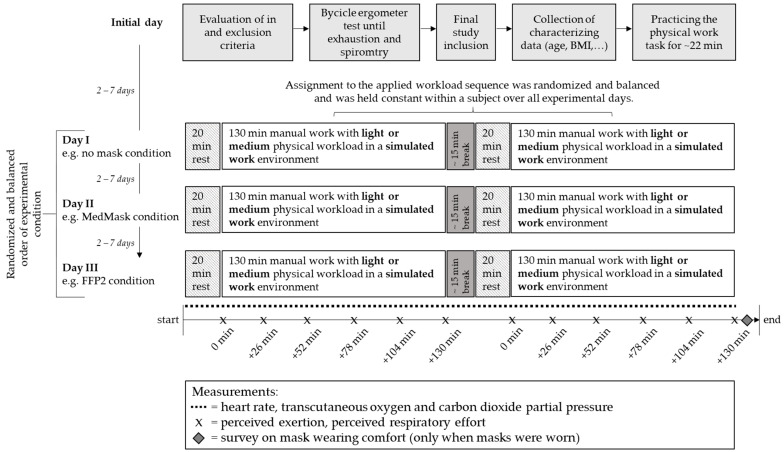
Study design. MedMask medical mask, FFP2 filtering face piece mask class II.

**Figure 2 healthcare-11-01308-f002:**
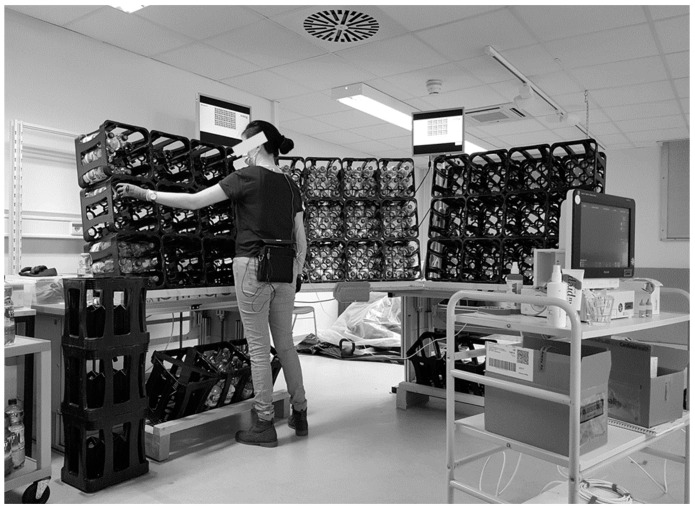
Simulated physical manual work task.

**Figure 3 healthcare-11-01308-f003:**
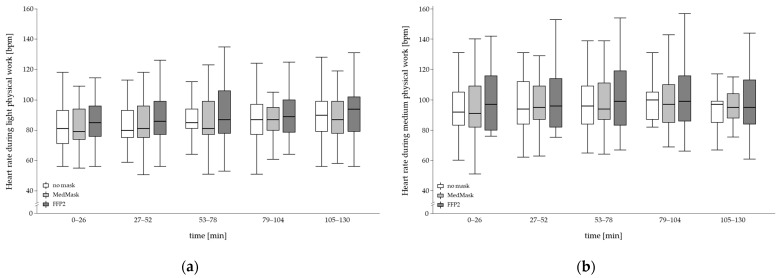
Temporal changes in heart rate during 130 min of light (**a**) and medium (**b**) physical manual work by mask condition (no mask, medical mask (MedMask) or filtering facepiece mask class II (FFP2)).

**Figure 4 healthcare-11-01308-f004:**
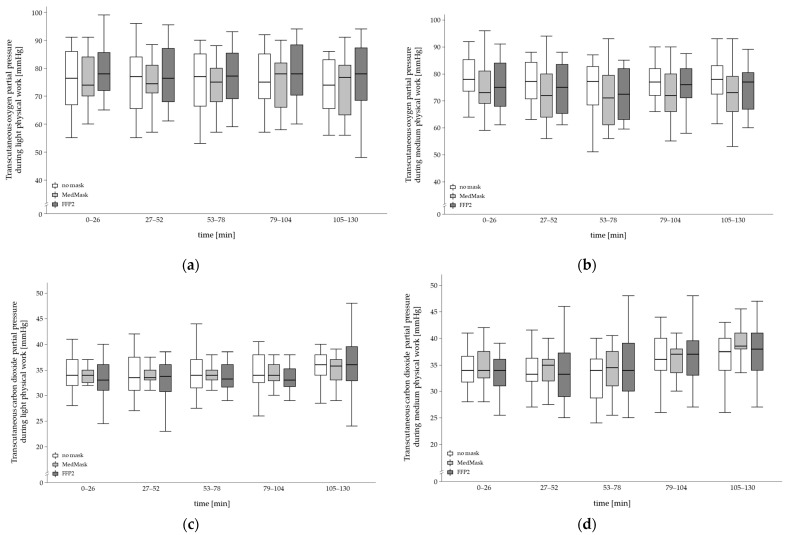
Temporal changes in transcutaneous oxygen (**a**,**b**) and carbon dioxide (**c**,**d**) partial pressure during 130 min of light (**a**,**c**) and medium (**b**,**d**) physical workload by mask condition (no mask, medical mask (MedMask) or filtering face piece mask class II (FFP2)).

**Figure 5 healthcare-11-01308-f005:**
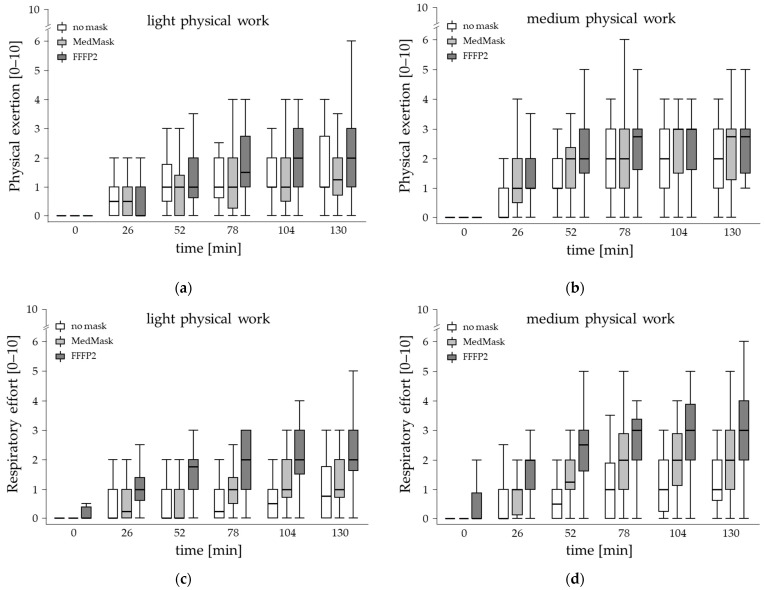
Temporal changes in perceived physical exertion (**a**,**b**) and respiratory effort (**c**,**d**) during light and medium physical manual work simulation by mask condition (no mask, medical mask (MedMask) or filtering facepiece mask class II (FFP2)).

**Figure 6 healthcare-11-01308-f006:**
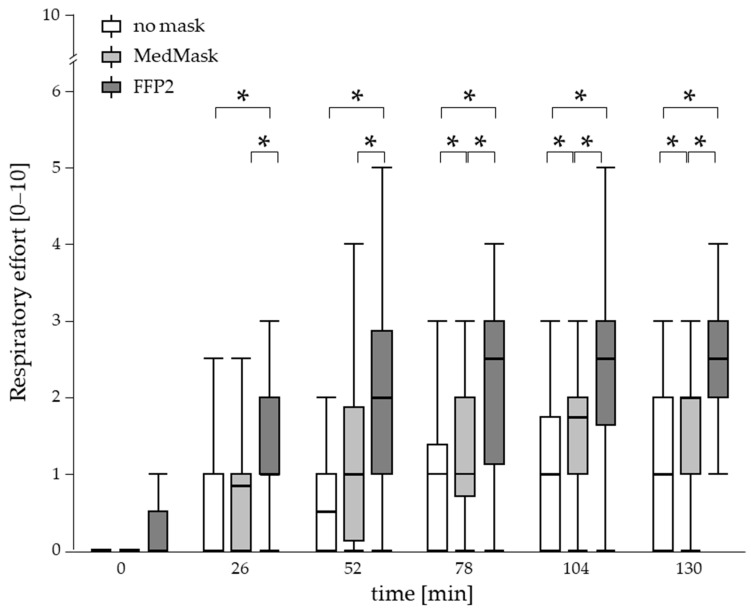
Perceived respiratory effort in the time course of 130 min simulated manual physical work by mask condition (no mask, medical mask (MedMask) or filtering facepiece mask class II (FFP2)) without differentiating for workload. Asterisks indicate statistically significant differences between the mask conditions (*p* < 0.01, adjusted for multiple comparison).

**Figure 7 healthcare-11-01308-f007:**
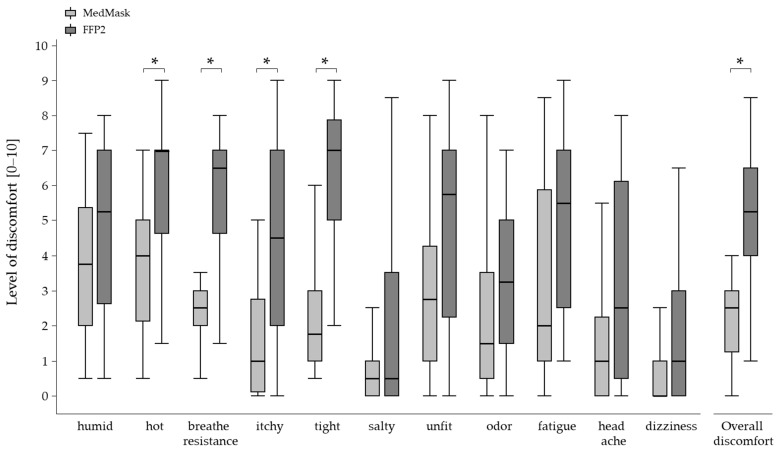
Differences in wearing comfort by mask condition (medical mask (MedMask) or filtering facepiece mask class II (FFP2)). Asterisks indicate statistically significant differences between the two mask conditions (*p* < 0.004, adjusted for multiple comparison).

**Table 1 healthcare-11-01308-t001:** Characteristics of the study sample.

Parameter			Overall	Fitness Level 1	Fitness Level 2	Fitness Level 3
PWCmax ^1^	[W/kg]	mean	3.1	2.5	3.0	4.4
		SD	0.8	0.4	0.2	0.8
sex	[n, %]	overall	24, 100%	9, 37.5%	10, 41.7%	5, 20.8%
		men	12, 50.0%	5, 20.8%	5, 20.8%	2, 8.3%
		women	12, 50.0%	4, 16.7%	5, 20.8%	3, 12.5%
smokers	[n, %]		1, 4.2%	1, 4.2%	0, 0%	0, 0%
age ^2^	[years]	mean	37.8	43.2	38.8	25.8
		SD	13.0	11.9	13.6	4.9
BMI ^2^	[kg/m²]	mean	23.8	25.1	23.3	22.3
		SD	2.3	2.1	2.3	1.5
Physical sport activity ^2, 3^	[minutes/week]	mean	132.6	56.1	128.5	278.3
		SD	147.3	106.7	131.7	152.6

PWCmax physical working capacity [mechanical power, Watt/kg] during a bicycle ergometer test until maximum exhaustion, BMI body mass index. ^1^ statistically significant difference between all three fitness levels, ^2^ statistically significant difference between fitness level 1 and 3, ^3^ statistically significant difference between fitness level 2 and 3.

**Table 2 healthcare-11-01308-t002:** Baseline values of heart rate, P_tc_O_2_, and P_tc_CO_2_ for each experimental day without wearing a mask.

Physiological Outcome	Statistics (F-Value *p*-Value)	Baseline Values before the Working Period and without Wearing a Mask (Means and Confidence Intervals)		
		No Mask	MedMask	FFP2
Heart rate [bpm]	F-value 0.59 *p* = 0.556	80.1 lower CI = 76.5 upper CI = 83.7	79.6 lower CI 76.0, upper CI 83.2	82.2 lower CI 78.6 upper CI 85.9
P_tc_O_2_ [mmHg]	F-value 1.89 *p* = 0.156	74.5 lower CI 70.5 upper CI 78.4	69.4 lower CI 65.4 upper CI 73.4	70.1 lower CI 66.0 upper CI 74.1
P_tc_CO_2_ [mmHg]	F-value 0.05 *p* = 0.954	32.4 lower CI 31.4 upper CI 33.	32.5 lower CI 31.5 upper CI 33.6	32.5 lower CI 31.5 upper CI 33.6

P_tc_O_2_ transcutaneous oxygen partial pressure, P_tc_CO_2_ transcutaneous carbon dioxide partial pressure, MedMask medical mask, FFP2 filtering face piece mask class II, CI confidence interval.

**Table 3 healthcare-11-01308-t003:** Linear mixed models for the physiological outcomes.

**Outcome**	**Factor**	**Degree of Freedom**	**F-Value**	***p*-Value**
heart rate	**mask condition**	2	**13.41**	**<0.0001**
	**workload**	1	**212.07**	**<0.0001**
	**time**	4	**6.71**	**<0.0001**
	fitness level	2	2.35	0.121
	mask condition × time	8	0.17	0.995
	mask condition × time × workload	8	0.20	0.990
	mask condition × time × fitness level	16	0.21	1.000
P_tc_O_2_	mask condition	2	4.09	0.017
	workload	1	0.63	0.429
	time	4	1.16	0.329
	fitness level	2	2.83	0.083
	mask condition × time	8	0.04	1.000
	mask condition × time × workload	8	0.03	1.000
	mask condition × time × fitness level	16	0.11	1.000
P_tc_CO_2_	mask condition	2	3.43	0.033
	**workload**	1	**16.88**	**0.0001**
	**time**	4	**26.17**	**<0.0001**
	fitness level	2	0.50	0.613
	mask condition × time	8	0.39	0.928
	mask condition × time × workload	8	0.62	0.760
	mask condition × time × fitness level	16	0.613	0.875

Bold letters indicate statistically significant model effects with *p* < 0.01 (adjusted for multiple comparison), P_tc_O_2_ transcutaneous oxygen partial pressure, P_tc_CO_2_ transcutaneous carbon dioxide partial pressure.

**Table 4 healthcare-11-01308-t004:** Linear mixed models for perceived exertion and respiratory effort.

Outcome	Factor	Degree of Freedom	F-Value	*p*-Value
perceived physical exertion	**mask condition**	2	**31.69**	**<0.0001**
	**workload**	1	**162.94**	**<0.0001**
	**time**	5	**153.36**	**<0.0001**
	fitness level	2	0.23	0.793
	mask condition × time	10	1.28	0.240
	mask condition × time × workload	10	0.68	0.739
	mask condition × time × fitness level	20	0.30	1.000
perceived respiratory effort	**mask condition**	2	**203.37**	**<0.0001**
	**workload**	1	**109.54**	**<0.0001**
	**time**	5	**103.86**	**<0.0001**
	fitness level	2	1.05	0.368
	**mask condition × time**	10	**6.68**	**<0.0001**
	mask condition × time × workload	10	0.31	0.978
	mask condition × time × fitness level	20	0.54	0.952

Bold letters indicate statistically significant model effects with *p* < 0.01 (adjusted for multiple comparison).

## Data Availability

The data are not publicly available due to data use restrictions contained in study participants’ information material.

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
