# Peer review of "Influence of Face Masks on Physiological and Subjective Response during 130 Min of Simulated Light and Medium Physical Manual Work—An Explorative Study"

_healthcare, 2023, doi:10.3390/healthcare11091308_

Round 1

Reviewer 1 Report

In my opinion, the study is interesting. The introduction should be developed and improved. The major weakness of the study is the small study group.  Please respond to my comments below.

L35 – ‘’ corona’’ - should be corrected to ''coronavirus''.

L39 – ‘’In 2023, the global COVID-19’’ - I would suggest expanding on the latest statistics on covidu, mortality, mutations, etc. It may be useful to work 10.1016/j.heliyon.2022.e08799

L49-59 - The authors write about the effects of masks on the human body, but their effect on the muscular system should be mentioned.  Information about the effect of masks on the muscular system of the masticatory muscles should be added in the introduction. The following works should be cited: 10.3390/jcm11020303 and 10.3390/ijerph192315559 .

L64 – ‘’ (for review see [11,15])’’ - think that the replacement with simply ''[11,15]'' will be better.

L72 – ‘’et al.’’ - The use of italics is not necessary.

L74-75 – ‘’ Georgi, Haase-Fielitz, Meretz, Gasert and Butter’’ - please provide only the first author.

L79-74 - Describe the purpose of the study clearly and add a research hypothesis.

L87 – ‘’Twenty-four healthy subjects (12 females, 12 males)’’ - Please clarify the following:

·       how did the authors calculate the sample size?

·       according to the criteria (L624), people of age 18 to 67 were included in the study. This is an awfully large age range. Why did the authors decide on one? Over the years, there are changes that can alter the body's vital parameters. This could have affected the outcome. Please explain.

L434 – ‘’ Table 4’’ - Is this an error? Is the text missing ?

4. Discussion - Note to all text. When authors cite studies and provide authors, quoting one name and adding et al. is sufficient. It is not necessary to give all the names. Example L531 – change ‘’ Li, Tokura, Guo, Wong, Wong, Chung and Newton’’ to ‘’Li et al. [27]’’.

L542 – ‘’ et al.’’ - The use of italics is not necessary.

Author Response

Dear reviewer,

Thank you very much for your time to critically read and comment on our manuscript. We have replied to your comments below and adjusted the manuscript’s text where necessary.

L35 – ‘’ corona’’ - should be corrected to ''coronavirus''.

  • We changed as suggested

L39 – ‘’In 2023, the global COVID-19’’ - I would suggest expanding on the latest statistics on covidu, mortality, mutations, etc. It may be useful to work 10.1016/j.heliyon.2022.e08799

  • our study focuses on a possible negative effect on workers by wearing face masks during physical work for infection prevention of Sars-Cov-2. We have therefore refrained from reporting in detail on cases, spread, mutation, treatment, etc.. However, see, that some more information about the whole pandemic situation would benefit to the manuscript as you suggested. We added a few statistics and general aspects of COVID-19 in the introduction (L39-44).

L49-59 - The authors write about the effects of masks on the human body, but their effect on the muscular system should be mentioned.  Information about the effect of masks on the muscular system of the masticatory muscles should be added in the introduction. The following works should be cited: 10.3390/jcm11020303 and 10.3390/ijerph192315559 .

  • We have reported possible negative effects on physiological vital signs. We did not address effects of masks on masticatory muscles in the introduction and would like to continue to refrain from doing so, as we do not contribute to this with our experimental data. However, it is true that this aspect should be mentioned in the manuscript since it is also a undesired side effect. We have now added some lines in the discussion (L632-636) and thank you for the valuable references.

L64 – ‘’ (for review see [11,15])’’ - think that the replacement with simply ''[11,15]'' will be better.

  • We changed as suggested

L72 – ‘’et al.’’ - The use of italics is not necessary.

  • We changed as suggested

L74-75 – ‘’ Georgi, Haase-Fielitz, Meretz, Gasert and Butter’’ - please provide only the first author.

  • We changed as suggested

L79-74 - Describe the purpose of the study clearly and add a research hypothesis.

  • We specified the research question of the study (L101). Since we conducted an exploratory study with a small number of subjects in a new research field, a specific directed hypothesis was not appropriate.

L87 – ‘’Twenty-four healthy subjects (12 females, 12 males)’’ - Please clarify the following:

  • how did the authors calculate the sample size?
  • The sample size is based on the amount of experimental conditions and was determined in order to avoid first order carry over effects. A sample size calculation based on a clinically relevant effect or no effect was not possible and we therefore consider our study as an explorative study knowing that the findings have to be verified by studies with larger sample sizes.
  • according to the criteria (L624), people of age 18 to 67 were included in the study. This is an awfully large age range. Why did the authors decide on one? Over the years, there are changes that can alter the body's vital parameters. This could have affected the outcome. Please explain.
  • Thank you for this comment. Including people of age 18 to 67 sounds really strange. However, our intention was to included people from the working age which is 18 to 67 in Germany. As we used an intra-individual study design (every subject was exposed to each experimental condition) changes in vital parameters due to aging may hardly play any role since “subject” was included as random factor to our statistical model. We changed the wording in line 683 to “People from the German Working Age (age between 18 and 67 years)”. Furthermore we amended the discussion about the sample size and sample characteristics in the limitation section (L625-627)

L434 – ‘’ Table 4’’ - Is this an error? Is the text missing ?

è Thank you, this was an error and we delete “table 4”.

  1. Discussion - Note to all text. When authors cite studies and provide authors, quoting one name and adding et al. is sufficient. It is not necessary to give all the names. Example L531 – change ‘’ Li, Tokura, Guo, Wong, Wong, Chung and Newton’’ to ‘’Li et al. [27]’’.
  • We changed as suggested

L542 – ‘’ et al.’’ - The use of italics is not necessary.

  • We changed as suggested

Reviewer 2 Report

General comments:

 The study is much needed because it explores parts of physical activity that are often not studied. Maximal sport and resting situations are often studied, but this study hits the nail on the head with the study of moderate intensity activities.

The main limitation of this work is that it talks about respiratory limitation with the use of different masks, but does not provide any measurement of ventilatory flow.

Some difficulties need to be explained by the authors in order to understand the study.

Why was 130 min chosen?

Specific comments

Introduction: The authors should discuss the resistance to ventilatory flow of each type of facemask and how this might influence the physiological response of people who are performing physical work.

A review of the literature on face masks in physical exercise would be necessary. In the same journal Int. J. Environ. Res. Public Health there are many articles worth reviewing.

Material and method: 

In participants: It should be indicated whether the phase of the menstrual cycle was controlled in women, or in which phase of the menstrual cycle the measurements were taken. In any case this should be discussed, as there is evidence that there may be differences in physiological response depending on the phase of the menstrual cycle.

Given that a maximal test has been performed, I would indicate the values obtained in the test and the protocol used. These values should serve as a reference to know how intense the 130 min effort was, compared to this test. At least in heart rate.

In Simulated light and moderate physical manual work, it would be important to know the intensity of the aforementioned tasks. For this purpose we recommend to use the table of Ainsworth 2000 (https://pubmed.ncbi.nlm.nih.gov/10993420/ ).

If the intensity of the tasks is moderate or low intensity, it is normal that there is no difference between the masks. When the tasks are of moderate or high intensity (> 6 MET's) you will start to find differences. This needs to be mentioned in the discussion.

Results:

It is necessary to include the flow diagram of the participants, where the whole process is observed, following the PRISMA flow diagram model.

Table two should be placed appropriately, as it is in the middle of the text.

It is clear that the intensity of the work is not well calibrated, because in both light and moderate work (figure 5 a and b) the mean values do not exceed 3 points in any case, which indicates that the effort was not really of medium intensity.

In line 433 there is a "Table 4" which I think has not been removed.

Discussion:

It should be indicated how intense the efforts were, comparing the maximal test with the heart rate of the activities. 

The issue of intensity of effort should be analysed in depth, comparing the proposed activity with a broad spectrum of activities, in order to see where similar activities might not make a difference. 

The duration should be discussed, as a person's working day is much longer than 130 min.

A clear limitation of this work that should be included in this section is that the possible lack of results is due to the fact that the intensity was not too high and this should be mentioned.

Author Response

Dear reviewer,

thank you for your assessment and comments. We have read your feedback carefully and tried to implement everything in the manuscript in an appropriate way. Thank you for the good advice.

General comments:

 The study is much needed because it explores parts of physical activity that are often not studied. Maximal sport and resting situations are often studied, but this study hits the nail on the head with the study of moderate intensity activities.

The main limitation of this work is that it talks about respiratory limitation with the use of different masks, but does not provide any measurement of ventilatory flow.

Some difficulties need to be explained by the authors in order to understand the study.

Why was 130 min chosen?

  • The reason for choosing an observation time of 130 min was added to the introduction (L66-73) and to the method section (L138-141)

Introduction: The authors should discuss the resistance to ventilatory flow of each type of facemask and how this might influence the physiological response of people who are performing physical work.

  • we adjusted the introduction to address this aspect (L57/58 and L62/63).

A review of the literature on face masks in physical exercise would be necessary. In the same journal Int. J. Environ. Res. Public Health there are many articles worth reviewing.

  • we have rewritten the introduction quite extensively and think that the focus could be put more clearly on the wearing of face masks during physical load during work and that the possible importance of different workload intensities has been addressed sufficiently (L93-98)

Material and method:

In participants: It should be indicated whether the phase of the menstrual cycle was controlled in women, or in which phase of the menstrual cycle the measurements were taken. In any case this should be discussed, as there is evidence that there may be differences in physiological response depending on the phase of the menstrual cycle.

  • The menstrual cycle was not monitored and not controlled in women. We believe that according to our intra-individual design with balanced randomization of the mask conditions physiological response differences related to the menstrual cycle can not have significantly affected our results. However, we cannot be sure and added a sentence to the limitation section (L630)

Given that a maximal test has been performed, I would indicate the values obtained in the test and the protocol used. These values should serve as a reference to know how intense the 130 min effort was, compared to this test. At least in heart rate.

  • we added this aspect to the limitation section (L610-615)

In Simulated light and moderate physical manual work, it would be important to know the intensity of the aforementioned tasks. For this purpose we recommend to use the table of Ainsworth 2000 (https://pubmed.ncbi.nlm.nih.gov/10993420/ ).

  • we added some information about the energy expenditure that is suggest for light and medium physical work (L616/617)

If the intensity of the tasks is moderate or low intensity, it is normal that there is no difference between the masks. When the tasks are of moderate or high intensity (> 6 MET's) you will start to find differences. This needs to be mentioned in the discussion.

  • We would like to respectfully challenge this assumption, as we do not find any relevant effects from wearing the face masks in our own work or in many other publications. And those papers that mention impaired physiology used spiroergometric evaluation which overestimates the effects of the face masks due to unrealistic leakage. However, we added the METs that are related to light and moderate physical manual work and mentioned that higher workloads may lead to different results (L617-621)

Results:

It is necessary to include the flow diagram of the participants, where the whole process is observed, following the PRISMA flow diagram model.

  • We used an intraindividual design and each subject completed each experimental condition. We think that Figure 1 shows exactly what you are referring to. We have looked again at the accompanying text and made some adjustments to express this more clearly (L149-151).

Table two should be placed appropriately, as it is in the middle of the text.

  • Actually table 2 is placed correctly at the end of subsection 3.4.1. Since this section is very short the layout looks a little bit odd.

It is clear that the intensity of the work is not well calibrated, because in both light and moderate work (figure 5 a and b) the mean values do not exceed 3 points in any case, which indicates that the effort was not really of medium intensity.

è The two activities were determined in accordance with a work design system REFA commonly used in industry to distinguish between light and medium physical work. However, we discussed that the range of physical workload maybe too low to find a moderating effect (L610-621).

In line 433 there is a "Table 4" which I think has not been removed.

è removed, thank you

Discussion:

It should be indicated how intense the efforts were, comparing the maximal test with the heart rate of the activities.

  • see discussion (L610-621).

The issue of intensity of effort should be analysed in depth, comparing the proposed activity with a broad spectrum of activities, in order to see where similar activities might not make a difference.

  • see discussion (L610-621).

The duration should be discussed, as a person's working day is much longer than 130 min.

  • we added this aspect to the limitation section of the discussion (L603)

A clear limitation of this work that should be included in this section is that the possible lack of results is due to the fact that the intensity was not too high and this should be mentioned.

  • see discussion (L610-621)

Reviewer 3 Report

This study aimed to investigate the influence of wearing a medical mask and a filtering facepiece class II respirator on the physiological and subjective outcomes in the course of manual work. The topic is interesting. Some comments for the authors to improve the quality of the manuscript are below.

1.       In the last paragraph of section 1, why did the 130 minutes were chosen as the experiment time? Is there any basis for this?

2.       In the section 2.2, why was heavy physical workload not considered?

3.       For the weight of containers, empty bottles, bottles filled with water or sand and water contents, etc. can preferably be stated separately.

4.       In section 3.2, why were smokers included as one of the parameters in the experiment?

5.       In section 3.4.1, “heart rate” had been repeated twice. Please correct.

6.       What research can the results of this study be used for? What are the directions and goals of future research? It can be added to the discussion.

Author Response

Dear Reviewer

Thank you for reviewing our manuscript and for your valuable comments. We have tried to implement everything adequately

This study aimed to investigate the influence of wearing a medical mask and a filtering facepiece class II respirator on the physiological and subjective outcomes in the course of manual work. The topic is interesting. Some comments for the authors to improve the quality of the manuscript are below.

  1. In the last paragraph of section 1, why did the 130 minutes were chosen as the experiment time? Is there any basis for this?
  • we added the reason for choosing 130 min in the introduction (L66-72) and in the methods as well (L138-141)
  1. In the section 2.2, why was heavy physical workload not considered?
  • Heavy physical work is less common than light and medium physical work. During the planning of the study, we decided not to use heavy or even the heaviest physical work as a further experimental condition in order not to increase the effort further. However, your comment has made us realize that this is a clear limitation of our study, so we have mentioned this accordingly in the limitation section (L610 - 621).
  1. For the weight of containers, empty bottles, bottles filled with water or sand and water contents, etc. can preferably be stated separately.
  • when describing the weights of loads to be handled, it is especially important to note that light physical work involves moving parts weighing 1kg and moderate physical work involves moving parts weighing up to 2kg, as this is the main differentiator between these work intensities. We therefore did not mention each part.
  1. In section 3.2, why were smokers included as one of the parameters in the experiment?
  • In section 3.2 we provide details about the characteristics of our final study sample. Smoking was no exclusion criteria and we wanted to report how many of the subjects considered themselves as smokers.
  1. In section 3.4.1, “heart rate” had been repeated twice. Please correct.
  • deleted, thank you!
  1. What research can the results of this study be used for? What are the directions and goals of future research? It can be added to the discussion.
  • thank you we added a final sentence to our conclusion section (L646-651)

Round 2

Reviewer 1 Report

The authors responded to my comments in a correct manner. I recommend the works for publication. With best regards.

Author Response

Dear reviewer,

thank you very much for your effort and feedback to imrpove our manuscript.

Reviewer 2 Report

We think the authors have done a good job in improving their publication. Sometimes, when you see the result of what you have done, it is easy to criticize the work of others and all reviewers can think of ways to improve the work, but it is undeniable that this work should be published and known for its direct practical application.

Author Response

Dear reviewer,

Thank you for your effort and valuable feedback to improve our manuscript.

Reviewer 3 Report

Most contents of this manuscript were modified more perfectly, need to pay attention to the chart and the text content alignment; In addition, the heavy physical work in the revised opinion in limitation is not very clear express, and it can be improved.

Author Response

Thank you for reviewing the revised manuscrict.

- We have included several references to the graphs and tables in the text to help the reader navigate the text with the associated tables and graphs.

- We rewrote the mentiones paragraph in the limitation section. "The work simulation also has some limitations, especially regarding the two applied physical workloads (light and medium workload). The workloads were determined according to REFA [28], a common method for the design of work systems. This determination of work load does not necessarily match with the physical activity level classification based on energy consumption according to metabolic equivalents. Since energy consumption was not measured in the present study, the heart rate analyses provide the best indication of whether the two work loads could be differentiated metabolically. The mean heart rate was 86 ± 16 bpm for the light physical workload and 95 ± 19 bpm for the medium physical workload. Expressed as relative values related to the maximum heart rate (HRmax) measured during the initial bicycle ergometer test, light physical workload was characterized by 49 ± 9 % HRmax and medium physical workload by 54 ± 11 % HRmax. These relative heart rates indicate that the difference in physical activity was rather small and both workloads were in in a low to very low intensity range [46]. Higher intensities such as those found in heavy physical work and heavy labor were not considered. Therefore, it could be possible that a moderating effect of workload was not detected due to a too narrow range of the investigated workloads."

- a spell check was conducted.